# Orthostatic Headache in Children Including Postural Tachycardia Syndrome and Orthostatic Hypotension: A Near-Infrared Spectroscopy Study

**DOI:** 10.3390/jcm9124125

**Published:** 2020-12-21

**Authors:** Soken Go, Gaku Yamanaka, Akiko Kasuga, Kanako Kanou, Tomoko Takamatsu, Mika Takeshita, Natsumi Morishita, Shinichiro Morichi, Yu Ishida, Shingo Oana, Yasuyo Kashiwagi, Takashi Mitsufuji, Nobuo Araki, Hisashi Kawashima

**Affiliations:** 1Department of Pediatrics and Adolescent Medicine, Tokyo Medical University, Tokyo 160-0023, Japan; gaku19710911@gmail.com (G.Y.); tea-party.alice@hotmail.co.jp (A.K.); kanako.hayashi.0110@gmail.com (K.K.); twomoonchild@gmail.com (T.T.); jerryfish_mika@yahoo.co.jp (M.T.); sunflowernk69@gmail.com (N.M.); shinichiro-m@hotmail.co.jp (S.M.); y.ishida31@gmail.com (Y.I.); shingooana@yahoo.co.jp (S.O.); hoyohoyo18@hotmail.com (Y.K.); hisashi@tokyo-med.ac.jp (H.K.); 2Department of Psychosomatic Medicine, Tokyo Metropolitan Children’s Medical Center, Tokyo 183-8561, Japan; 3Department of Neurology, Saitama Medical University, Saitama 350-0495, Japan; mitsufuj@saitama-med.ac.jp (T.M.); arakin@saitama-med.ac.jp (N.A.)

**Keywords:** head and/or neck pain, the International Classification of Headache Disorders 3rd edition, children, cerebral blood flow fluctuation, migraine, postural tachycardia syndrome, pediatric, adolescent, headache, orthostatic dysregulation, orthostatic intolerance, orthostatic headache, cerebrospinal fluid leak, active standing test, head up-tilt

## Abstract

Background and aim: Although head and/or neck pain attributed to orthostatic hypotension is included in international guidelines, its mechanisms and relevance remain unknown. This study examined the term’s relevance and aimed to elucidate the associated clinical features. Methods: An active stand test was performed to evaluate fluctuations in systemic and cerebral circulation in children and adolescents reporting complaints in the absence of a confirmed organic disorder. The subjects were categorized based on orthostatic headache presence/absence, and their characteristics and test results were compared. Results: Postural tachycardia syndrome was observed in 50.0% of children with, and 55.1% without, orthostatic headache. For orthostatic hypotension, the respective values were 31.3% and 30.6%. A history of migraine was more prevalent in children with orthostatic headaches (64.1% vs. 28.6%; *p* < 0.01). The observed decrease in the cerebral oxygenated hemoglobin level was larger in children with orthostatic headaches (Left: 6.3 (3.2–9.4) vs. 4.1 (0.8–6.1); *p* < 0.01, Right: 5.3 (3.1–8.6) vs. 4.0 (0.8–5.9); *p* < 0.01). Conclusion: Fluctuations in cerebral blood flow were associated with orthostatic headaches in children, suggesting that the headaches are due to impaired intracranial homeostasis. As orthostatic headache can have multiple causes, the term “head and/or neck pain attributed to orthostatic (postural) hypotension” should be replaced with a more inclusive term.

## 1. Introduction

Headaches are common in childhood and adolescence, with approximately 20% of children reporting having a headache at least once a year [1]. In children, recurrent headaches are predominantly primary headaches, such as migraine or tension-type headaches.

Chronic headaches may greatly interfere with daily life. Children with orthostatic headaches experience difficulties in daily life. However, their condition is difficult to treat because, being induced by postural changes such as standing, it does not meet the diagnostic criteria for either primary or secondary headaches. When moving from the supine to a standing position, the volume of blood that returns to the heart decreases by approximately 30%, lowering cardiac output and systemic blood pressure and triggering the baroreceptor reflex. Although this reflex maintains adequate, blood pressure in healthy people, those with an abnormal baroreceptor reflex or decreased plasma volume experience a severe drop in blood pressure upon standing. In adults, orthostatic hypotension (OH) is diagnosed when a decrease in systolic blood pressure ≥ 20 mmHg or diastolic blood pressure ≥ 10 mmHg occurs within 3 min of changing from the supine or sitting to the standing position. 

Low et al. published a study of postural tachycardia syndrome (POTS) in 1995 [2]. POTS commonly effects women in their 40s, and many patients first notice their symptoms in childhood or adolescence [3]. The current global diagnostic criteria for POTS in children and adolescents include marked tachycardia (>40 bpm) triggered by 10 min of active standing after rising from a supine position or a head-up tilt test. The criterion for POTS is a lack of a marked decrease in blood pressure. However, like OH, POTS results from abnormal changes in blood circulation mediated by the autonomic nervous system when assuming a standing position. Thus, POTS is often considered to be similar to OH. As a result that POTS involves orthostatic tachycardia associated with orthostatic symptoms, it is thought to encompass multiple pathological conditions with such symptoms as severe and/or long-lasting fatigue, light-headedness with prolonged sitting or standing, brain fog, forceful heartbeats or palpitations, nausea, abdominal pain, and headache.

A disorder termed “head and/or neck pain attributed to orthostatic (postural) hypotension” was included in the third edition of the International Classification of Headache Disorders (ICHD3). This concept, which has yet to become established, is defined as “pain, mostly in the back of the neck but sometimes spreading upwards to the occipital region (“coat hanger” distribution), attributed to postural hypotension presenting only in upright posture” (ICHD3, A 10.7) [4]. Impaired intracranial homeostasis has been suggested as a possible cause. However, this concept is derived from studies on OH due to neurological diseases in the elderly, such as Parkinson’s disease, multisystem atrophy, and trauma [5,6], and it is unclear whether it applies to orthostatic headaches in children and adolescents. Moreover, it is still unclear whether POTS without OH is consistent with this disease concept.

The present study examined the relevance of the term, “head and/or neck pain attributed to orthostatic (postural) hypertension,” as included in the ICHD3 by investigating the mechanisms of orthostatic headaches in children based on clinical features and cerebral blood flow fluctuations.

## 2. Methods

### 2.1. Participants

Pediatric patients who visited our hospital between January 2017 and March 2019 due to three or more concurrent complaints disrupting daily life and school attendance and lasting more than two months, such as headache, malaise, palpitations, morning fatigue, daily nausea, face pallor, umbilical colic, and orthostatic dizziness, were identified. Patients with an obvious, posttraumatic headache were excluded at the initial visit. The eligible patients underwent an examination which included detailed history taking, neurophysical tests, thoracoabdominal radiography, head magnetic resonance imaging (MRI), echocardiography, 12-lead electrocardiography, and blood analysis, including thyroid hormone measurement. If no clearly organic disorder was found, the subjects and their parents were asked to submit their verbal and written informed consent to participate. All participating subjects underwent the active stand test (described below), and their cerebral blood flow was simultaneously assessed. None of the subjects received any drug affecting their blood pressure or modifying their NE level, such as selective serotonin reuptake inhibitor (SSRI) or atomoxetine, before the test. The participants were divided into those who either did or did not experience a headache induced by the active stand test (the orthostatic headache vs. non-orthostatic headache groups). The groups were compared in terms of their characteristics, background factors, and systemic and cerebral circulatory responses. In addition, the association between the presence/absence of pulsatility in induced headaches and headache sites and other factors were examined.

### 2.2. Patient Characteristics and Background

The patient characteristics of age, sex, height, weight, body mass index (BMI), and cardiothoracic ratio (CTR) were examined. The background factors assessed included the presence/absence of migraine and morning fatigue. Migraine was diagnosed based on the ICHD3 criteria.

### 2.3. Active Stand Test

The subjects were admitted on the day before the active stand test. They underwent the test before breakfast on the following day between 8:00 and 9:30 a.m. in a quiet room with an ambient temperature between 23–24 degrees Celsius and no external light. After resting in the supine position for 10 min, they were asked to stand up in 3 s and remain standing for 10 min. Subsequently, they were asked to return to the supine position and rest for 5 min. Readings were taken continuously throughout this procedure. Blood pressure, pulse, and blood pressure recovery time were measured using a continuous noninvasive pressure system (Finometer MIDI, Finapres Medical Systems) with a small to medium-sized probe placed on the left middle finger (or another finger, depending on the size of their fingers). To determine the timing of subjective symptom onset, the subjects were asked by the examiner whether they had a headache before standing and at every minute until the conclusion of the test. None of the subjects had a headache while in the supine position. Their responses were recorded. After the conclusion of the test, the subjects were asked about the site and features of their headache, if any, and the presence or absence of pulsatility. If the subject felt severe orthostatic intolerance or presyncope symptoms, the test was terminated, and the point was recorded, but there were only few such instances. In addition, to measure the blood norepinephrine (NE) level, an indwelling needle was inserted and fixed to the left cubital vein at least 30 min before the start of the active stand test. Flashback blood samples were collected before, and 10 min after, standing. POTS was diagnosed if a sustained heart rate increase of 40 bpm within 10 min of standing was recorded in the absence of OH.

### 2.4. Evaluation of Cerebral Blood Flow

The probes of an infrared oximeter (NIRO-200 NX, Hamamatsu Photonics) were attached to the left and right sides of the frontal region, and a hairband was used to fix the probes and prevent light leakage. Highly permeable light at 700 to 900 nm was used, and the light scattered in the body was automatically measured with a detector. The oxygenated hemoglobin (OxyHb) concentrations in the bilateral frontal cerebral cortices were measured using the Modified Beer–Lambert method [7]. To evaluate the change in OxyHb concentration from immediately before to after standing, the OxyHb level at the time of headache onset in the orthostatic headache group and the lowest OxyHb level while standing in the non-orthostatic headache group were analyzed. Determining if there was any correlation between the OxyHb and clinical symptoms was impossible because the blood pressure recovery time immediately after standing was very brief. Thus, the OxyHb levels obtained during the recovery time were not included in the analysis.

### 2.5. Statistical Analysis

SPSS 25.0 (SPSS Inc., Chicago, IL, USA) was used to perform the Mann–Whitney and chi-squared tests. *p* < 0.01 was considered to indicate statistical significance.

### 2.6. Ethical Approval and Details of Study Protocol Registration

Written informed consent was obtained from all the subjects for the publication of the details of their case. A copy of their written consent is available for review upon request. All procedures involving human participants were done in accordance with the ethical standards of the institutional and national research committees (Ethics Committee of the Tokyo Medical University; SH3337) and the 1964 Helsinki Declaration and its later amendments or comparable ethical standards. As this study is a clinical audit of ICHD3 criteria, it does not necessarily require ethics approval.

## 3. Results

The inclusion criteria were met by 113 subjects who subsequently underwent the active stand test, including 64 in the orthostatic headache group and 49 in the non-orthostatic headache group (Figure 1). POTS was found in 32 subjects (50.0%) in the orthostatic headache group, and 27 subjects (55.1%) in the non-orthostatic headache group. OH was found in 20 (31.3%) and 15 (30.6%) subjects in the orthostatic headache and non-orthostatic headache groups, respectively. No statistically significant difference was observed between groups (*p* = 0.876).

Similarly, the subject characteristics, including age, sex, body constitution, and CTR, did not differ significantly between the groups. Although no significant intergroup difference was observed in the background factors, there were significantly more subjects with a history of migraine in the orthostatic headache group than in the non-orthostatic headache group (64.1% vs. 28.6%; *p* = 0.001) (Table 1). No significant intergroup differences were observed in the systolic or diastolic blood pressure, heart rate, blood NE level in the supine position at rest, blood pressure recovery time, maximum heart rate, heart rate increment or NE level upon standing. However, a more substantial decrease in cerebral OxyHb level on the left and right sides after standing was observed in the orthostatic headache group (Left: 6.3 [3.2–9.4] vs. 4.1 [0.8–6.1], *p* = 0.006; Right: 5.3 [3.1–8.6] vs. 4.0 [0.8–5.9], *p* = 0.007) than in the non-orthostatic headache group.

Of the 11 somatic symptoms of orthostatic dysregulation (OD) as defined by the Japanese Society of Psychosomatic Pediatrics [8], only headache was experienced most often in the daily life of the subjects in the orthostatic headache group. However, no significant intergroup difference was observed in any other symptoms, including dizziness on standing, fainting in a standing position, nausea on taking a hot bath, palpitations after mild exercise, morning fatigue, face pallor, anorexia, umbilical colic, fatigability, and carsickness (Table 2).

The most common site of headache upon standing was the whole head (17 [26.6%]), followed by the temporal region (16 [25%]) and the frontal or occipital region (11 [17.2%]). In 18 of the 64 (28%) children with orthostatic headaches, the headaches induced by the active stand test were pulsatile. No statistical difference was observed in the site of headaches between subjects with and without pulsatile orthostatic headaches (*p* = 0.214) (Table 3A). Neither the proportion of subjects with a high NE level (*p* = 0.493) nor the prevalence of concomitant migraine (*p* = 0.441) differed between the groups with and without a pulsatile headache (Table 3B). All subjects with an orthostatic headache reported an improvement in their symptom within 5 min of assuming the supine position after the test.

## 4. Discussion

Orthostatic headaches have traditionally been associated with intracranial hypotension. In 2003, Mokri et al. reported that many patients with orthostatic headaches did not, in fact, exhibit features of cerebrospinal fluid leak; moreover, many of them received the diagnosis of POTS following an evaluation of their autonomic nervous system [9]. As a result that the subjects were children, examinations requiring a spinal tap, such as cisternal scintigraphy and magnetic resonance myelography, were not performed to avoid exposing participants unnecessarily to risks associated with these invasive procedures. However, no evidence of intracranial hypotension, such as subdural effusion, dilated epidural venous plexus or diffuse dural enhancement, was observed in the subjects with chronic headaches or episodes of orthostatic headache who had undergone an MRI. In both groups, the median age was almost the same. The gender distribution was slightly skewed toward females. In terms of body constitution, the subjects in both groups were somewhat thin (BMI: 18.7 and 19.3, respectively), with an average BMI < 22, which is considered the cutoff value for normal weight. These findings were consistent with the epidemiological evidence of OD in Japan [8]. Previous studies have also shown that BMI tends to be lower in children with POTS than in healthy children [10], a finding also in line with those of the present study.

### 4.1. “Head and/or Neck Pain Attributed to Orthostatic (Postural) Hypotension”—Concept Relevance

In the present study, 64 of the 113 subjects experienced an orthostatic headache. However, the proportion of normal subjects and those with POTS or OH did not differ significantly between the groups with and without orthostatic headaches. Previously, Deb et al. reported that 87% of patients with POTS experienced headaches in daily life [11]. Moreover, Khurana et al. reported that among patients with POTS, 58.3% and 62.5% developed orthostatic headaches in daily life and during the head-up tilt test, respectively. They also reported that among patients aged ≤ 30 years, the risk of orthostatic headaches increased the longer they remained in the standing position [12]. In the present study, the 10-min active stand test, which is the standard technique for evaluating autonomic function in children in Japan, induced an orthostatic headache in 32 of the 59 subjects with POTS (54.2%), which is comparable with previously reported data [12]. It is noteworthy that the previously reported results were reproduced in a laboratory setting. Furthermore, orthostatic headaches occurred in children with POTS without hypotension, such as those with OH; the present study demonstrated that the incidence of orthostatic headaches was higher than expected. Previous studies on vasovagal syncope reported a decrease in the mean blood flow velocity in the middle cerebral artery and an increase in cerebrovascular resistance despite the occurrence of syncope [13,14]. A separate study reported that when presyncope occurred before the onset of deceleration or hypotension, reversible electroencephalographic abnormalities with discrepancies between the right and left signals were observed at the time of syncope aura occurrence [15]. It is plausible that in POTS, in circumstances that cause an abnormal increase in cerebrovascular resistance, failure of tachycardia to compensate for poor blood reflux from the lower extremities or a mild decrease in blood pressure causing a rapid decrease in cerebral blood flow might be experienced as headache, consistent with presyncope symptoms. Our study revealed that the right and left cerebral OxyHb levels were significantly lower in the orthostatic headache group than in the non-orthostatic headache group. Kim et al., who performed the active stand test in children in the general population without an underlying disease using the same device as in our study, reported that the standing posture did not induce any symptoms in subjects with an OxyHb decrease ≤ 4 μM/L in contrast to subjects with a decrease ≥ 6.4 μM/L, who reported symptoms of OI [16]. These values are comparable with those obtained in our study. Fujita et al. evaluated the cerebral blood flow separately in children with POTS, INOH, and delayed OH and reported that changes in cerebral blood flow with a discrepancy between the right and left sides preceded changes in blood pressure and heart rate [17,18,19]. The lack of any difference in the proportion of children with POTS and OH between the orthostatic headache and non-orthostatic headache groups in our study indicated that orthostatic headaches are a pathological condition that might depend on impaired intracranial homeostasis, which therefore cannot be confirmed by readings of superficial fluctuations in blood pressure and heart rate alone. The concept of “head and/or neck pain attributed to orthostatic (postural) hypotension” included in the ICHD3 should not be restricted by terminology describing superficial changes in the systemic circulation, such as OH and POTS. Instead, a more inclusive term reflecting the wide variety of possible etiologies should be adopted.

### 4.2. Orthostatic Headaches and Norepinephrine

Thieben et al. reported no association between excretion of urinary sodium (Na), which reflects the circulating blood volume, and increase in noradrenaline level, which reflects sympathetic nervous activity after standing [3]. The authors suggested that failure of the compensatory mechanism for hypovolemia due to peripheral venous pooling or marked leakage from the capillaries in the lower extremities after standing may cause orthostatic headaches.

Meanwhile, the renin-angiotensin-aldosterone system is an important system for maintaining the circulating blood volume. A previous study reported low renin levels in patients with POTS despite hypovolemia [20], drawing attention to insufficient Na retention due to abnormalities in this system as a possible cause of hypovolemia. Although the CTR greatly varies among individuals and does not always accurately reflect the left ventricular volume or circulating blood volume, it is often used as a simple indicator in clinical practice. The lack of apparent differences in CTR between the subjects with and without orthostatic headaches in our study indicated that orthostatic headaches might not be caused by changes in the plasma volume, which are reflected in the CTR. Although the normal CTR in children ranged from 41–50%, in the present study, the median CTR was 41% and 42% for the orthostatic headache and non-orthostatic headache groups, respectively, suggesting that the subjects in our study with complaints had certain factors which were reflected by low CTR. Low et al. reported that NE levels rose markedly (≥600 pg/mL) upon standing in one-third of his subjects [3,21] and defined hyperadrenergic POTS as an “increase in systolic blood pressure by 10 mmHg or more, and NE level of 600 pg/mL or higher after standing.” Although Zhang et al. reported that the incidence of headache, dizziness, and tremulousness was higher among patients with hyperadrenergic POTS than in those with other types of POTS [22], our study revealed no significant change in the NE level before or after standing between subjects with and without orthostatic headaches. In addition, no association was observed between the pulsatility of orthostatic headaches and blood NE levels upon standing. However, these issues may require further research with a larger number of subjects.

### 4.3. Orthostatic Headaches and Migraine

Ojha et al. reported that the incidence of concomitant migraine in patients with POTS was 41% in adolescents and 61% in adult patients [23]. Likewise, Raj et al. reported that 41% of patients with POTS had a concomitant migraine [24]. Meanwhile, Khurana reported that orthostatic headache in patients with POTS was characterized by moderate-to-severe, migraine-like, pulsatile headaches affecting the whole head or frontal region which were frequently accompanied by photosensitivity, hyperacusis, and nausea. However, the orthostatic headaches tended to be less persistent than migraines in many subjects and were able to be relieved in more than 50% of the subjects by remaining in the supine position for 2–15 min [12]. In our study, all subjects with orthostatic headache reported an improvement in their symptoms within 5 min of assuming the supine position, and pulsatility was observed in 18 of the 64 subjects (28.1%) in the orthostatic headache group. Furthermore, the subjects in the orthostatic headache group reported headaches more frequently in their daily life than those in the non-orthostatic headache group while the prevalence of migraine was higher in the orthostatic headache group. Given that the orthostatic headache group included 32 subjects with POTS, accounting for half the enrolled subjects, in addition to 20 children with OH, their experience of headache induced by standing might have been misdiagnosed as a migraine by their attending clinicians, who may have relied on headache duration described in the diagnostic criteria for migraine. Miglis reported that migraine with an aura tended to cause more pronounced autonomic dysfunction than migraine without an aura; a higher likelihood of impairment of the sympathetic, rather than the parasympathetic, nervous system; severer sympathetic nervous impairment during the interictal period of a migraine; and increased sympathetic nervous responsiveness during the ictal period, presumably due to adrenoreceptor hypersensitivity [25]. A review article by Gazerani et al. showed a complex pattern of altered autonomic nervous system function associated with migraine, but with an imbalance between the sympathetic and parasympathetic nervous systems [26]. On the other hand, Novak et al. reported that the pathogenesis of POTS appears to involve a hyperadrenergic state and distal neuropathy affecting sympathetic myocardial fibers [27]. Pengo et al. reported that patients with POTS showed neither polysomnographic findings consistent with associated sleep pathology nor objective daytime sleepiness, but subjective daytime sleepiness was associated with increased activation of the parasympathetic nervous system [28]. Gibbons et al. reported cholinergic POTS patients had abnormal proximal sudomotor function and symptoms suggesting gastrointestinal and genitourinary parasympathetic nervous system dysfunction [29]. These may suggest an association between migraine and POTS as an autonomic dysfunction. The present study did not examine the role of the aura, its features, its association with other factors or sympathetic and parasympathetic nervous function; instead, it explored potential differences between typical migraine and orthostatic headaches based on the site of the induced headache, pulsatility, and NE levels upon standing, among other considerations. However, no clear trends were found. At present, there is no consensus on treatment strategies for headaches associated with abnormal intracranial homeostasis. Theoretically, common analgesics and triptan are likely to be ineffective [30]. However, the similarity in the clinical features between the orthostatic headaches in our patients and POTS may provide a clue to their treatment.

### 4.4. Limitations

The limitation of this study is that the symptoms, aura, nausea, and photosensitivity associated with orthostatic headache were not assessed, and the association with migraine was not fully investigated. More research is needed on these points in the future.

## 5. Conclusions

Our study found that orthostatic headaches in children were associated with fluctuations in cerebral blood flow and were induced by impaired intracranial homeostasis. In some children, the features of orthostatic headaches were found to be similar to those of migraine. However, in almost all the cases, the headache improved upon assuming the supine position sooner than described in the criteria for migraine diagnosis. Special attention should be paid in clinical practice to this similarity, as the treatment can differ. As orthostatic headaches occur in patients with OH and POTS, the description, “head and/or neck pain attributed to orthostatic (postural) hypotension” in the ICHD3 should be replaced with a more inclusive, diagnostic term that better defines the condition or differentiates between disparate entities on the basis of head/neck pain attributed to orthostatic hypotension and orthostatic headache associated with impaired intracranial vascular homeostasis. Reconceptualizing orthostatic headaches may help clinicians find more effective treatments and contribute to progress in the research on the disease.

## Figures and Tables

**Figure 1 jcm-09-04125-f001:**
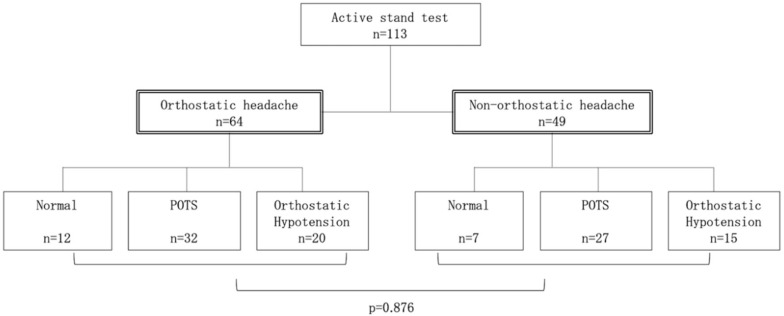
Categorization of participants, orthostatic headache, and autonomic disorders. The inclusion criteria were met by 113 subjects who subsequently underwent the active stand test, including 64 in the orthostatic headache group and 49 in the non-orthostatic headache group. Postural tachycardia syndrome (POTS) was found in 32 subjects (50.0%) in the orthostatic headache group, and 27 subjects (55.1%) in the non-orthostatic headache group. Orthostatic hypotension was found in 20 (31.3%) and 15 (30.6%) subjects in the orthostatic headache and non-orthostatic headache groups, respectively. No statistically significant difference was observed between groups (*p* = 0.876). Values indicate the number of subjects. Statistical analysis was performed using the chi-squared test. *p* < 0.01 was considered to indicate statistical significance.

**Table 1 jcm-09-04125-t001:** Participant characteristics and standing test results.

		Orthostatic Headache (*n* = 64)	Non-Orthostatic Headache (*n* = 49)	*p*-Value
Profile and Background	Age (years)	14.0 (13.2–15.1)	14.4 (13.2–15.5)	0.491
	Male ratio (%)	31/64 (48.4)	24/49 (48.9)	0.954
	Height (cm)	158.9 (152.9–164.5)	159.0 (152.6–166.2)	0.823
	Weight (kg)	49.7 (40.3–55.3)	49.6 (43.7–55.0)	0.956
	BMI	18.7 (16.9–20.9)	19.3 (17.7–20.1)	0.876
	CTR (%)	42 (38–44)	41 (38–44)	0.363
	Migraine (%)	64.1	28.6	0.001 *
Supine	SBP (mmHg)	108 (101–120)	107 (99–117)	0.496
	DBP (mmHg)	60 (51–67)	54 (48–63)	0.096
	HR (bpm)	66 (58–80)	68 (64–80)	0.295
	Norepinephrine (pg/mL)	153 (129–225)	160 (121–214)	0.842
Standing	Recovery time (sec)	20 (16–22)	18 (15–20)	0.099
	Maximum HR (bpm)	111 (93–125)	111 (97–123)	0.768
	HR increase (bpm)	38 (28–51)	38 (29–50)	0.871
	Lt cerebral oxyHb decrease (μmol/L)	6.3 (3.2–9.4)	4.1 (0.8–6.1)	0.006 *
	Rt cerebral oxyHb decrease (μmol/L)	5.3 (3.1–8.6)	4.0 (0.8–5.9)	0.007 *
	Norepinephrine (pg/mL)	352 (253–466)	328 (252–453)	0.623

The subject characteristics, including age, sex, body constitution, and CTR, did not differ significantly between the groups. Although no significant intergroup difference was observed in the background factors, there were significantly more subjects with a history of migraine in the orthostatic headache group than in the non-orthostatic headache group (64.1% vs. 28.6%; *p* = 0.001). A more substantial decrease in cerebral OxyHb level on the left and right sides after standing was observed in the orthostatic headache group (Left: 6.3 [3.2–9.4] vs. 4.1 [0.8–6.1], *p* =0.006; Right: 5.3 [3.1–8.6] vs. 4.0 [0.8–5.9], *p* = 0.007) than in the non-orthostatic headache group. Values are expressed as the median (IQR) or %. BMI = body mass index, CTR = cardiothoracic ratio, SBP = systolic blood pressure, DBP = diastolic blood pressure, HR = heart rate, Rt = right, Lt = left. Statistical analysis was performed using the Mann–Whitney test. * *p* < 0.01 was considered to indicate statistical significance.

**Table 2 jcm-09-04125-t002:** Daily physical symptoms.

	Orthostatic Headache (*n* = 64)			Non-Orthostatic Headache (*n* = 49)			*p*-Value
	Frequently	Sometimes	Rarely	Frequently	Sometimes	Rarely	
1. Dizziness on standing	55	5	4	36	8	5	0.156
2. Fainting in the standing position	33	19	12	23	16	10	0.843
3. Nausea on taking a hot bath	28	16	20	14	15	20	0.227
4. Palpitations after mild exercise	31	13	20	16	15	18	0.171
5. Morning fatigue	55	7	2	40	4	5	0.310
6. Face pallor	17	20	27	21	8	20	0.119
7. Anorexia	20	13	31	10	10	29	0.394
8. Umbilical colic	28	18	18	17	11	21	0.287
9. Fatigability	47	11	6	38	8	3	0.806
10. Headache	53	8	3	28	9	12	0.004 *
11. Car sickness	33	15	16	24	15	10	0.575

Of the 11 somatic symptoms of orthostatic dysregulation (OD) as defined by the Japanese Society of Psychosomatic Pediatrics, headache was experienced most often in the daily life of the subjects in the orthostatic headache group. However, no significant intergroup difference was observed in any of the other symptoms. Values indicate the number of subjects. Statistical analysis was performed using the chi-squared test. * *p* < 0.01 was considered to indicate statistical significance.

**Table 3 jcm-09-04125-t003:** Pulsatility of orthostatic headache.

(**A**)								
	Whole head	Frontal	Parietal	Temporal	Posterior	Indeterminate		Total
Pulsatility (+)	5	2	0	8	3	0		18
Pulsatility (−)	12	9	4	8	8	5		46
Total	17	11	4	16	11	5		64
*p* = 0.214								
(**B**)								
	NE ≥ 600 pg/mL		NE < 600 pg/mL		Migraine (+)		Migraine (−)	
Pulsatility (+)	3		15		12		6	
Pulsatility (−)	5		41		29		17	
Total	8		56		41		23	
	*p* = 0.493				*p* = 0.441			

The most common site of headache upon standing was the whole head (17 [26.6%]), followed by the temporal region (16 [25%]) and the frontal or occipital region (11 [17.2%]). In 18 of the 64 (28%) children with orthostatic headaches, the headaches induced by the active stand test were pulsatile. No statistical difference was observed in the site of headaches between subjects with and without pulsatile orthostatic headache (*p* = 0.214) (**A**). Neither the proportion of subjects with a high NE level (*p* = 0.493) nor the prevalence of concomitant migraine (*p* = 0.441) differed between the groups with and without a pulsatile headache (**B**). Values are number of subjects. Statistical analysis was performed by the Chi-squared tests. *p* < 0.01 was considered to indicate statistical significance.

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
