# Peer review of "Orthostatic Headache in Children Including Postural Tachycardia Syndrome and Orthostatic Hypotension: A Near-Infrared Spectroscopy Study"

_jcm, 2020, doi:10.3390/jcm9124125_

Round 1

Reviewer 1 Report

I think the research is well designed and these results suggest that an impairment of intracranial circulatory homeostasis could play a role not only in patients with POTS or OH, but also in patients without systemic autonomic disorders. The study includes a numerous sample of patients, the active stand test is well protocolized and the infrared oximeter method has been proven to be reliable in other studies. The results are also well discussed and interpreted.

I have several recommendations and requests:

  • The inclusion criteria seem unspecific (i.e. “multiple complaints”). Please specify the patient profile and symptoms that were considered for the inclusion in the study. Please state in methods if you took into account exclusion criteria such as any underlying diseases or previous medications that could have influenced the results (i.e. BP lowering drugs, antidepressants or other drugs that modify NE levels, etc.).
  • In case of medications, were they suspended before the active stand test?
  • During the test, please specify if syncope or other orthostatic intolerance symptoms were recorded.
  • I find the orthostatic headache is poorly described. If it is possible and recorded, it would be interesting to know other clinical characteristics: unilateral/bilateral, duration, resolution with supine position, the presence of photophobia, phonophobia, osmophobia, sensibility to movement, nausea/vomiting. In patients with previous migraine, does it resemble migraine attacks?
  • In relation to this, one question to be better defined in the discussion would be if orthostatic mechanisms in this series of patients could be a trigger for episodes of migraine (at least in patients with previous migraine) or depicts an independent entity itself. The authors express that orthostatic headache might have been misdiagnosed as migraine by their attending clinicians. But it could be the opposite situation. Migraine is not always pulsatile and the location can be variable, so this is not sufficient to answer this question.
  • I agree that orthostatic headache is a controversial entity, poorly defined and that merits a reconceptualization in the ICHD-3. I suggest, but leave it to the authors discretion, not to demand the term to be replaced with a more inclusive term, but perhaps differentiate different entities (i.e. head/neck pain attributed to orthostatic hypotension and orthostatic headache associated with impaired intracranial vascular homeostasis).

Author Response

Dear reviewer 1

Thank you for your constructive review of our manuscript. We also appreciate your advice for improving our manuscript.

We carefully revised our manuscript to address the concerns raised. The responses are given below.

We hope that our responses are satisfactory, and that the manuscript is now suitable for publication.

Thank you again for considering our revised manuscript. We look forward to hearing from you.

Sincerely yours,

Reviewer's Comments to the authors:

  • The inclusion criteria seem unspecific (i.e. “multiple complaints”). Please specify the patient profile and symptoms that were considered for the inclusion in the study. Please state in methods if you took into account exclusion criteria such as any underlying diseases or previous medications that could have influenced the results (i.e. BP lowering drugs, antidepressants or other drugs that modify NE levels, etc.).
  • In case of medications, were they suspended before the active stand test?

Response:

Thank you for calling our attention to these important points. We have made the following corrections with some explanations as shown below.

(Page 2)

Pediatric patients who visited our hospital between January 2017 and March 2019 due to three or more concurrent complaints disrupting daily life and school attendance and lasting more than two months, such as headache, malaise, palpitations, morning fatigue, daily nausea, face pallor, umbilical colic, and orthostatic dizziness, were identified.

None of the subjects received any drug affecting their blood pressure or modifying their NE level, such as SSRI or atomoxetine, before the test.

  • During the test, please specify if syncope or other orthostatic intolerance symptoms were recorded.

Response:

We are grateful for making this important point. We made the following change.

(Page 3)

 If the subject felt severe orthostatic intolerance or presyncope symptoms, the test was terminated, and the point was recorded, but there were only few such instances.

  • I find the orthostatic headache is poorly described. If it is possible and recorded, it would be interesting to know other clinical characteristics: unilateral/bilateral, duration, resolution with supine position, the presence of photophobia, phonophobia, osmophobia, sensibility to movement, nausea/vomiting. In patients with previous migraine, does it resemble migraine attacks?

Response:

We are grateful for making this important point. Unfortunately, this study did not examine photophobia, osmophobia, sensibility to movement or nausea/vomiting. To the headaches listed in Table 3, we added that all subjects improved within 5 minutes of lying down.

(Page 6)

All subjects with an orthostatic headache reported an improvement in their symptom within 5 minutes of assuming the supine position after the test.

  • In relation to this, one question to be better defined in the discussion would be if orthostatic mechanisms in this series of patients could be a trigger for episodes of migraine (at least in patients with previous migraine) or depicts an independent entity itself. The authors express that orthostatic headache might have been misdiagnosed as migraine by their attending clinicians. But it could be the opposite situation. Migraine is not always pulsatile and the location can be variable, so this is not sufficient to answer this question.

Response:

We appreciate your valuable observations. Indeed, migraines are not characterized solely by location or pulsatility. However, in our study, all the subjects with orthostatic headache with some migraine characteristics improved within 5 minutes after assuming the supine position, suggesting that misdiagnosis may have occurred due to reliance on the headache duration in the diagnostic criteria for migraine headache. Thus, we added the following clarification.

(Page 9)

In our study, all subjects with orthostatic headache reported an improvement in their symptoms within 5 minutes of assuming the supine position, and pulsatility was observed in 18 of the 64 subjects (28.1%) in the orthostatic headache group.

Given that the orthostatic headache group included 32 subjects with POTS, accounting for half the enrolled subjects, in addition to 20 children with OH, their experience of headache induced by standing might have been misdiagnosed as a migraine by their attending clinicians, who may have relied on headache duration described in the diagnostic criteria for migraine.

Our study found that orthostatic headaches in children were associated with fluctuations in cerebral blood flow and were induced by impaired intracranial homeostasis. In some children, the features of orthostatic headaches were found to be similar to those of migraine. However, in almost all the cases, the headache improved upon assuming the supine position sooner than described in the criteria for migraine diagnosis.

  • I agree that orthostatic headache is a controversial entity, poorly defined and that merits a reconceptualization in the ICHD-3. I suggest, but leave it to the authors discretion, not to demand the term to be replaced with a more inclusive term, but perhaps differentiate different entities (i.e. head/neck pain attributed to orthostatic hypotension and orthostatic headache associated with impaired intracranial vascular homeostasis).

Response:

We appreciate your valuable recommendation. We have added the following.

(Page 10)

As orthostatic headaches occur in patients with OH and POTS, the description, “head and/or neck pain attributed to orthostatic (postural) hypotension” in the ICHD3 should be replaced with a more inclusive, diagnostic term that better defines the condition or differentiates between disparate entities on the basis of head/neck pain attributed to orthostatic hypotension and orthostatic headache associated with impaired intracranial vascular homeostasis.

Reviewer 2 Report

General comments

  1. This is a fascinating topic and idea for investigation. I commend the authors for undertaking the project
  2. The writing seems to be a bit blocky instead of flowing from one thought to the next. The authors may benefit from professional proofreading. 

Abstract

  1. The methods section does not clearly explain which patient populations were being compared

Introduction

  1. On page 2, the first full paragraph jumps between ideas stating that the criteria for POTS is a lack of decreased BP, and only later does it mention that the heart rate increases and the value that is diagnostic. 
  2. I would recommend discussing the changes seen in the brain on fMRI in POTS and how this may mediate headache in the introduction as this may help the reader understand the mechanisms of the interaction a bit better

Methods

  1. Was this project approved by an IRB committee or oversight board?
  2. Were these patients who presented to an outpatient clinic or emergency room?  If a clinic, what type of clinic?
  3. I am a bit confused with what the authors mean when they state "no organic disorder found", and then look at organic disorders such as POTS.
  4. The fact that patients with post-traumatic headaches were excluded should be mentioned here rather than in the discussion.  Any other relevant inclusion or exclusion criteria should be clearly delineated.
  5. So there were patients without any headache at all?  It seems the patient pool included those with a variety of chief complaints but from the description it is not clear if all had some form of headache (even if not orthostatic).
  6. Why was irritable bowel syndrome diagnosed? This hasn't been mentioned in the paper up to this point and doesn't seem to be clearly related to POTS or migraines in this context.
  7. Were the patients given a standard fluid regimen or have fluids restricted prior to the test?

Results

  1. You mention 133 subjects, but figure 1 only shows 113.  Was this a typo or are there other patients without headache?
  2. Did the induced orthostatic pulsatile headaches have any other migrainous features?
  3. It should be mentioned in the results rather than discussion that there were no indications of intracranial hypotension on workup (that as I understand it, all patients underwent)
  4. What was the rate of change in cerebral blood flow?  That may be another interesting piece of data rather than just the delta.

Discussion- overall this section seems disorganized and fails to present coherent thoughts, but does bring up interesting points.

  1. "It is noteworthy that the previously
    reported results were reproduced in a laboratory setting"- do you mean this study when you refer to a laboratory setting?
  2. There is also evidence of parasympathetic dysfunction in migraine that would be worth commenting on in the context of your results

Conclusion

  1. You mention that special attention should be paid to migrainous features of orthostatic headache patients, but it is unclear why you say this or how it would change treatment
  2. What would you recommend as an example of a better ICHD-3 term?

Author Response

Dear reviewer 2

Thank you for your constructive review of our manuscript. We also appreciate your advice for improving our manuscript.

We carefully revised our manuscript to address the concerns raised. The responses are given below.

We hope that our responses are satisfactory, and that the manuscript is now suitable for publication.

Thank you again for considering our revised manuscript. We look forward to hearing from you.

Sincerely yours,

Reviewer's Comments to the authors:

  • On page 2, the first full paragraph jumps between ideas stating that the criteria for POTS is a lack of decreased BP, and only later does it mention that the heart rate increases and the value that is diagnostic.

Response:

Thank you for calling our attention to these important points. We made the following corrections with some explanations for clarification.

(Page 2)

Low et al. published a study of postural tachycardia syndrome (POTS) in 1995.2 POTS commonly affects women in their 40’s, and many patients first notice their symptoms in childhood or adolescence.3 The current global diagnostic criteria for POTS in children and adolescents include marked tachycardia (>40 bpm) triggered by 10 minutes of active standing after rising from a supine position or a head-up tilt test. The criterion for POTS is a lack of a marked decrease in blood pressure. However, like OH, POTS results from abnormal changes in blood circulation mediated by the autonomic nervous system when assuming a standing position. Thus, POTS is often considered to be similar to OH. Because POTS involves orthostatic tachycardia associated with orthostatic symptoms, it is thought to encompass multiple pathological conditions with such symptoms as severe and/or long-lasting fatigue, light-headedness with prolonged sitting or standing, brain fog, forceful heartbeats or palpitations, nausea, abdominal pain, and headache. 

  • I would recommend discussing the changes seen in the brain on fMRI in POTS and how this may mediate headache in the introduction as this may help the reader understand the mechanisms of the interaction a bit better

Response:

We appreciate your valuable recommendation but can’t find the article about fMRI in POTS.

  • Was this project approved by an IRB committee or oversight board?

Response:

Yes, it was. We added this to Methods as shown below.

(Page 3/4)

2.6. Ethical approval

Written informed consent was obtained from all the subjects for the publication of the details of their case. A copy of their written consent is available for review upon request.

2.7. Details of study protocol registration

All procedures involving human participants were done in accordance with the ethical standards of the institutional and national research committees (Ethics Committee of the Tokyo Medical University; SH3337) and the 1964 Helsinki Declaration and its later amendments or comparable ethical standards.

  • Were these patients who presented to an outpatient clinic or emergency room?  If a clinic, what type of clinic?

Response:

This study was conducted in a specialized outpatient department by appointment only at an university hospital.

  • I am a bit confused with what the authors mean when they state "no organic disorder found", and then look at organic disorders such as POTS.

Response:

We appreciate your observation. We corrected the text as shown below.

(Page 3)

If no clearly organic disorder was found, the subjects and their parents were asked to submit their verbal and written informed consent to participate.

  • The fact that patients with post-traumatic headaches were excluded should be mentioned here rather than in the discussion.  Any other relevant inclusion or exclusion criteria should be clearly delineated.

Response:

Thank you for your important comments. We added to Methods that post-trauma patients were excluded.

  • So there were patients without any headache at all?  It seems the patient pool included those with a variety of chief complaints but from the description it is not clear if all had some form of headache (even if not orthostatic).

Response:

Thank you for your important comment. As shown in Table 2, there were a few subjects who did not have headache.

  • Why was irritable bowel syndrome diagnosed? This hasn't been mentioned in the paper up to this point and doesn't seem to be clearly related to POTS or migraines in this context.

Response:

We appreciate your questions. We removed IBS as it was not important to this study.

  • Were the patients given a standard fluid regimen or have fluids restricted prior to the test?

Response:

Thank you for your question. After the diagnosis of POTS or OH, adequate fluid intake was encouraged, but no specific fluid intake instructions were given for the days before the examination. However, on the morning of the examination, drinking water was restricted.

  • You mention 133 subjects, but figure 1 only shows 113.  Was this a typo or are there other patients without headache?

Response:

Thank you for pointing this out. This is a typo.

  • Did the induced orthostatic pulsatile headaches have any other migrainous features?

Response:

Thank you for making this important point. Unfortunately, this study was unable to examine photophobia, osmophobia, sensibility to movement or nausea/vomiting. To the headaches listed in Table 3, we added that all subjects improved within 5 minutes of lying down.

(Page 6)

All subjects with an orthostatic headache reported an improvement in their symptom within 5 minutes of assuming the supine position after the test.

  • What was the rate of change in cerebral blood flow?  That may be another interesting piece of data rather than just the delta.

Response:

Thank you for your observation. The value of cerebral blood flow listed in Table 1(cerebral oxyHb decrease) shows the rate of change.

  • "It is noteworthy that the previously
    reported results were reproduced in a laboratory setting"- do you mean this study when you refer to a laboratory setting?

Response:

Yes, this study showed that the structured method gave reproducible results.

  • There is also evidence of parasympathetic dysfunction in migraine that would be worth commenting on in the context of your results

Response:

Thank you for your important comment. We added an explanation, citing articles related to migraine and POTS parasympathetic dysfunctions as shown below.

(Page9)

A review article by Gazerani et al. showed a complex pattern of altered autonomic nervous system function associated with migraine, but with an imbalance between the sympathetic and parasympathetic nervous systems27. On the other hand, Novak et al. reported that the pathogenesis of POTS appears to involve a hyperadrenergic state and distal neuropathy affecting sympathetic myocardial fibers28. Pengo et al. reported that patients with POTS showed neither polysomnographic findings consistent with associated sleep pathology nor objective daytime sleepiness, but subjective daytime sleepiness was associated with increased activation of the parasympathetic nervous system29. And Gibbons et al, reported cholinergic POTS patients had abnormal proximal sudomotor function and symptoms suggesting gastrointestinal and genitourinary parasympathetic nervous system dysfunction30. These may suggest an association between migraine and POTS as an autonomic dysfunction.

  • You mention that special attention should be paid to migrainous features of orthostatic headache patients, but it is unclear why you say this or how it would change treatment

Response:

Although the clinical similarities between orthostatic headache and migraine are confusing, we think it is important to pay attention to this point because there is a difference in the time till improvement after assuming the supine position especially, which may lead to different treatment choices. We have added the following.

(Page10)

In some children, the features of orthostatic headaches were found to be similar to those of migraine. However, in almost all the cases, the headache improved upon assuming the supine position sooner than described in the criteria for migraine diagnosis. Special attention should be paid in clinical practice to this similarity, as the treatment can differ.

  • What would you recommend as an example of a better ICHD-3 term?

Response:

Thank you for your important comment. For example, I think of it as "cranial homeostasis-related headache," but please forgive me for not mentioning this in the article.

Reviewer 3 Report

The authors present a very interest study on a very relevant topic. I have included some comments below that I feel would benefit the presentation and clarity of the project.

Please provide much more information on statistics. For example, if the chi-square test was performed, what was the output value? Only the p-value was provided. Also unclear what the statistics were performed on - e.g., table 3 - was this on total scores or the + or - pulsatility

Table captions would greatly benefit the reader in terms of understanding the contents of each table.

Table 1 - I am confused as to how a percentage can be statistically significant than a different percentage. This is in relation to migraine percentage and irritable bowel syndrome. 

Table 3 - there is a symbol under 'Pulsatility (-)' that I am assuming based on the other table should reflect 'total'

It would be nice to see correction for the number of statistical tests performed. Also, to be more transparent than p<0.01. It is not clear this would survive a full correction and providing real p values would assist the reader in determining this. 

I did not appreciate a limitations sections. There of course limitations that should be considered with any investigation and it would be nice to see authors address these in a structured format.

Author Response

Dear reviewer 3

Thank you for your constructive review of our manuscript. We also appreciate your advice for improving our manuscript.

We carefully revised our manuscript to address the concerns raised. The responses are given below.

We hope that our responses are satisfactory, and that the manuscript is now suitable for publication.

Thank you again for considering our revised manuscript. We look forward to hearing from you.

Sincerely yours,

Reviewer's Comments to the authors:

  • Please provide much more information on statistics. For example, if the chi-square test was performed, what was the output value? Only the p-value was provided. Also unclear what the statistics were performed on - e.g., table 3 - was this on total scores or the + or – pulsatility
  • Table captions would greatly benefit the reader in terms of understanding the contents of each table.

Response:

Thank you for calling our attention to these important points. We add the captions and statical information to the table and figure.

  • Table 1 - I am confused as to how a percentage can be statistically significant than a different percentage. This is in relation to migraine percentage and irritable bowel syndrome. 

Response:

We appreciate your valuable recommendation. IBS was removed as it is not important to this study.

  • Table 3 - there is a symbol under 'Pulsatility (-)' that I am assuming based on the other table should reflect 'total'

Response:

We appreciate your valuable comment and have made the recommended correction.

  • It would be nice to see correction for the number of statistical tests performed. Also, to be more transparent than p<0.01. It is not clear this would survive a full correction and providing real p values would assist the reader in determining this.

Response:

We appreciate your valuable comments and have made the recommended correction.

  • I did not appreciate a limitations sections. There of course limitations that should be considered with any investigation and it would be nice to see authors address these in a structured format.

Response:

We appreciate your important comment and have added the limitations as shown below.

(Page 10)

The limitation of this study is that the symptoms, aura, nausea, and photosensitivity associated with orthostatic headache were not assessed, and the association with migraine was not fully investigated. More research is needed on these points in the future.

Round 2

Reviewer 2 Report

Great job with the revision! The paper referenced below is to what I was referring earlier.  I do not necessarily think it needs to be included in your paper after the other changes you made, though, as you now have a better argument for why orthostatic changes and migraines may be linked.

Umeda S, Harrison NA, Gray MA, Mathias CJ, Critchley HD. Structural brain abnormalities in postural tachycardia syndrome: A VBM-DARTEL study. Front Neurosci. 2015;9:34. Published 2015 Mar 17. doi:10.3389/fnins.2015.00034
